# Performance of Polystyrene-Impregnated and CCA-Preserved Tropical Woods Against Subterranean Termites in PNG Field and Treatment-Induced Color Change

**DOI:** 10.3390/polym17141945

**Published:** 2025-07-16

**Authors:** Yusuf Sudo Hadi, Cossey Yosi, Paul Marai, Mahdi Mubarok, Imam Busyra Abdillah, Rohmah Pari, Gustan Pari, Abdus Syukur, Lukmanul Hakim Zaini, Dede Hermawan, Jingjing Liao

**Affiliations:** 1Department of Forest Products, Faculty of Forestry and Environment, IPB University (Bogor Agricultural University), Kampus IPB Darmaga, Bogor 16680, Indonesia; yshadi@indo.net.id (Y.S.H.); ibusyra.a@gmail.com (I.B.A.); abdssykr18@gmail.com (A.S.); lukmanhz@apps.ipb.ac.id (L.H.Z.); dedehe@apps.ipb.ac.id (D.H.); 2School of Forestry, Faculty of Natural Resources, Papua New Guinea University of Technology, Taraka Campus, Lae 411, Papua New Guinea; cossey.yosi@pnguot.ac.pg; 3Forest Products Programme, Forest Research Institute, Lae 411, Papua New Guinea; pmaraii37@gmail.com; 4National Research and Innovation Agency (BRIN), Cibinong 16911, Indonesia; rohm017@brin.go.id (R.P.); gustanpari@gmail.com (G.P.); 5College of Landscape and Horticulture, Yunnan Agricultural University, Kunming 650201, China

**Keywords:** polystyrene wood, CCA-preserved wood, discoloration, termite resistance, tropical wood

## Abstract

Logs supplied in Papua New Guinea and Indonesia are predominantly sourced from fast-growing tree species of plantation forests. The timber primarily consists of sapwood, which is highly susceptible to biodeterioration. At a training center, CCA (chromated copper arsenate) is still used for wood preservation, while in the wood industry, ACQ (alkaline copper quaternary) is commonly applied to enhance the service life of timber. In the future, polystyrene impregnation or other non-biocidal treatments could potentially serve this purpose. This study aimed to determine the discoloration and resistance of polystyrene-impregnated and CCA-preserved woods. Wood samples, *Anisoptera thurifera* and *Octomeles sumatrana* from Papua New Guinea, and *Anthocephalus cadamba* and *Falcataria moluccana* from Indonesia, were used. The wood samples were treated with polystyrene impregnation, CCA preservation, or left untreated, then exposed at the PNG Forest Research Institute site for four months. After treatment, the color change in polystyrene-impregnated wood was minor, whereas CCA-preserved wood exhibited a noticeably different color compared to untreated wood. The average polymer loading for polystyrene-impregnated wood reached 147%, while the average CCA retention was 8.4 kg/m^3^. Densities of untreated-, polystyrene-, and CCA-wood were 0.42, 0.64, and 0.45 g/cm^3^, respectively, and moisture contents were 15.8%, 9.4%, and 13.4%, respectively. CCA preservation proved highly effective in preventing termite attacks; however, CCA is hazardous to living organisms, including humans. Polystyrene impregnation also significantly improved wood resistance to subterranean termites, as indicated by lower weight loss and a higher protection level compared to untreated wood. Additionally, polystyrene treatment is nonhazardous and safe for living organisms, making it a promising option for enhancing wood resistance to termite attacks in the future as an alternative to the biocides currently in use.

## 1. Introduction

Wood remains one of the most widely used for construction and structural materials worldwide, particularly in tropical regions due to its natural abundance, renewability, and desirable mechanical properties [1]. However, despite these advantages, some tropical woods are highly vulnerable to biological deterioration, including fungal decay and, particularly, insect infestation, which significantly reduces their durability and service life [2]. Among wood-destroying insects, subterranean termites pose a particularly severe threat in tropical climates, causing extensive structural damage and considerable economic losses every year [3,4].

Conventional methods for wood protection, such as the application of chemical preservatives containing heavy metals or synthetic biocides, have been widely used to enhance the durability and service life of timber products in various ecosystems [5,6,7]. However, in Papua New Guinea, for wood preservation purposes, CCA (chromated copper arsenate) is still used at a training center, while ACQ is applied in industries. These poisonous/harmful chemicals for wood preservation raise environmental and health concerns, including chemical leaching into soil and groundwater, as well as toxicity risks to humans and ecosystems. Consequently, there is a growing global demand for environmentally friendly wood preservation strategies that effectively reduce biodegradation while minimizing ecological impacts [8,9,10].

As alternative solutions to these growing environmental concerns, wood modifications, including thermal treatments, impregnation, and chemical modifications, have been increasingly studied and industrially implemented [8,9,10,11,12,13,14,15,16]. In particular, wood chemical modifications such as furfurylation or acetylation are considered capable of improving wood properties, specifically its dimensional stability and durability against biodeterioration, including its resistance to termite attacks. However, these modifications generally require a higher investment, especially in the quantity of the reactive chemicals used and the recovery of the post-impregnating solutions [14].

Based on the above limitations, in this study, the impregnation of wood using styrene monomer can be another alternative to improve wood properties, particularly its durability against biodeterioration. The recovery issue of the post-impregnating solution can be eliminated since the styrene monomer is relatively stable and can be reused many times. Polymer impregnation technology, particularly using polystyrene, has emerged as a promising approach to improve the durability and resistance of wood against biological attacks [17]. Polystyrene impregnation can significantly reduce moisture uptake, enhance dimensional stability, and decrease susceptibility to biodegradation agents [18]. Despite these advantages, limited research has focused specifically on the application and performance of polystyrene impregnation for tropical wood species, especially under actual field conditions where termite pressure is high [19,20]. Therefore, to further investigate the termite resistance of polystyrene-modified wood under real field conditions, a series of treatments, including impregnation with a well-known wood preservative, should be conducted and evaluated [21]. In addition, although polystyrene-modified wood is not a new technology, the application of this technique to certain wood species from Papua New Guinea (PNG), which are relatively new in the field of wood modification, offers valuable insights for readers, particularly for PNG citizens who wish to enhance the durability and stability of their native wood resources.

The main objective of this study was to evaluate the effectiveness of polystyrene impregnation and compare it to the currently used preservative on four selected tropical wood species in terms of resistance to subterranean termite attack under natural field exposure conditions in Papua New Guinea. Specifically, this study aimed to assess whether polystyrene-impregnated tropical woods demonstrate significantly enhanced resistance to termite infestation compared to untreated controls, thus providing an environmentally friendly alternative for timber preservation in tropical climates.

## 2. Materials and Methods

### 2.1. Materials

Two wood species from Papua New Guinea, namely *Anisoptera thurifera* (density: 0.6–0.65 g/cm^3^) [22] and *Octomeles sumatrana* (density: 0.25–0.38 g/cm^3^) [23], were obtained from a timber shop in East Taraka, Lae, Papua New Guinea. The other two wood species, *Anthocephalus cadamba* (density: 0.29–0.56 g/cm^3^) [24] and *Falcataria moluccana* (density: 0.23–0.50 g/cm^3^) [25], were sourced from a timber shop in Bogor, Indonesia. The four wood species were cut into test specimens measuring 0.8 cm in thickness, 2 cm in width, and 20 cm in length, then dried in an oven at 60 °C until their moisture content reached approximately 12%. The chemicals used were CCA (chromated copper arsenate), obtained from the Timber and Forestry Training College at PNG University of Technology in Lae, Papua New Guinea. Styrene monomers (purity: ≥99%) and potassium peroxydisulfate (purity: ≥99%) (used as a catalyst) were purchased from Toko FRP Jakarta and PT. Merck Indonesia Tbk, Jakarta, Indonesia, respectively.

### 2.2. Wood Treatment Process

The test sample of each wood species was placed in a tank and subjected to a vacuum of 600 mmHg for 30 min. They were then submerged in a styrene monomer containing a catalyst (potassium peroxydisulfate; 2% *w*/*w*) and subjected to a pressure of 10 kg/cm^2^ for 30 min [18]. The wood samples were left immersed in styrene monomer for 18 h. After impregnation, the samples were removed from the vacuum pressure apparatus and placed on a rack to drain excess impregnating solution.

The impregnated wood samples were wrapped in aluminum foil and subjected to a polymerization process at 100 °C for 24 h. The weight percentage gain (WPG) was calculated as the percentage increase in oven-dry weight of the wood after treatment, relative to the oven-dry weight before treatment [20], using the following formula:WPG (%) = [(W_1_ − W_0_)/W_0_] × 100
where W_0_ is the initial oven-dry weight, and W_1_ is the oven-dry weight after impregnation and curing.

For comparison, untreated (control) and CCA-preserved wood specimens were also prepared. For CCA preservation, a 3% CCA solution was used in the same vacuum-pressure treatment as described for polystyrene-impregnated wood. All the wood samples were measured for density and moisture content according to British Standard (BS) 373:1957 [26].

### 2.3. Discoloration

The processes of polystyrene impregnation and CCA preservation in the wood may cause color changes or discoloration in the wood. The color and discoloration of all wood specimens were determined using the CIELab method by measuring L* (lightness), a* (red to green), and b* (blue to yellow) values [27], with a scanner (CanoScan 4400F) and Adobe Photoshop CS5. The color change (ΔE) of wood specimens was also calculated based on the CIELab method, while classification followed Hunter Lab and Hrčková et al. [28,29].

### 2.4. Termite Field Test

After the conditioning process, the samples were weighed, and their moisture content was measured. The samples were then exposed in the field at the Forest Research Institute in Lae City, Papua New Guinea (6°43′11.7″ S, 146°59′49.4″ E). The test site conditions are shown in Figure 1.

The samples were stacked vertically with 15 cm buried in the ground, maintaining a spacing of approximately 40 cm between each sample for a period of 4 months. The average temperature and annual rainfall in Lae, Papua New Guinea, are presented in Figure 2, with an annual rainfall of 6797 mm, an average temperature of 23.6 °C, and an elevation of 29 m above sea level.

At the end of the test, the wood samples were removed, cleaned of soil and debris, and then dried to reach oven-dry weight. The samples were weighed and examined to determine the protection level according to ASTM D 1758-06 [31]. The weight loss (%) of each sample after the field test was calculated using the following formula:Weight loss (%) = [(W_2_ − W_1_)/W_1_] × 100 
where W_1_ and W_2_ are the oven-dried weights of the samples before and after the field tests, respectively.

### 2.5. Scanning Electron Microscopy (SEM) Analysis

To confirm the penetration of polystyrene into the wood voids, a scanning electron microscope (SEM Hitachi SU3500, Tokyo, Japan) was used to detect the presence of polystyrene. During the preparation of wood samples for SEM analysis, the specimens are first oven-dried at approximately 60 °C. This drying step aims to eliminate moisture that could interfere with imaging under vacuum conditions. Once dried, the samples are cut into small pieces, roughly 5 mm × 5 mm× 5 mm in size, typically using a microtome or a sharp blade. The cuts are made in either the transverse direction. The sample surface was sputter-coated with a thin gold layer before observation.

After the samples are placed in the SEM chamber, imaging parameters such as accelerating voltage (between 10 and 20 kV) and working distance (around 10 mm) are adjusted to achieve optimal image resolution. Observations begin at low magnification to locate regions of interest, followed by gradually increasing the magnification to capture detailed microstructural features, such as vessels, fibers, and lumens. These images enable the detection of physical changes resulting from treatment, including the presence of polystyrene deposits or alterations in cell wall structure.

### 2.6. Energy Dispersive X-Ray Spectroscopy (EDS) Analysis

Following SEM imaging, EDS analysis was conducted to determine the elemental composition of specific regions within the wood samples. Using the EDS detector integrated with the SEM system, selected areas, such as fiber walls, vessel lumens, or filled voids, were analyzed through area mapping. For wood treated with polystyrene, the key elements of interest were carbon (C) and oxygen (O). Polystyrene-treated samples generally exhibit a stronger carbon signal compared to untreated wood, which naturally has a high oxygen content due to its polymeric structure. Elemental distribution maps were generated to visualize the spatial dispersion of these elements within the wood structure. These maps are typically color-coded, for example, cyan for carbon and green for oxygen, to enhance interpretability. All spectral data, element maps, and associated SEM images should be carefully archived for further analysis and for comparing treated samples with control specimens.

### 2.7. Data Analysis

This study consists of two factors. The first factor is wood species, with four levels: *Anisoptera thurifera*, *Octomeles sumatrana*, *Anthocephalus cadamba*, and *Falcataria moluccana*. The second factor is wood treatment, with three levels: untreated wood, polystyrene-impregnated wood, and CCA-preserved wood. Each combination of wood species and treatment was replicated eight times, resulting in a total of 96 samples (four wood species × three treatments × eight replicates).

The experiment followed a completely randomized block design and can be expressed linearly as follows:y = u + b + t + ε
where y is the response, u is the overall mean, b is the block effect, t is the treatment effect, and ε is the experimental error.

Analysis of variance (ANOVA) was performed. If any factor showed a significant difference at a *p* ≤ 0.05, further analysis was conducted using Duncan’s multiple-range test [18].

## 3. Results and Discussion

### 3.1. Physical Properties

Physical properties in this study cover wood discoloration, density, and moisture content. These topics are discussed as follows.

#### 3.1.1. Wood Discoloration

To increase the service life of wood, preservation could be applied by introducing chemical preservatives into the wood or by impregnating it with styrene, followed by a polymerization process [19,32,33]. Both techniques could enhance the Ewood’s resistance to biodeterioration attacks. However, these treatments may cause changes in wood color. As shown in Figure 3, the color and condition of all wood samples, untreated, polystyrene-impregnated, and CCA-preserved woods, are displayed before field exposure. The color parameters (L*, a*, b* system) and color change (ΔE) of these wood samples are presented in Table 1.

From Table 1, it can be observed that the L*, a*, and b* values of untreated and polystyrene-impregnated wood are similar or show only slight changes, while CCA wood shows significantly different values from both, indicating that discoloration has occurred in the CCA wood. For further data analysis, Table 2 presents a summary of the variance analysis of all measured parameters in this study.

Table 2 shows that L*, a*, b*, and ∆E are significantly affected by wood species and treatment factors. For further data analysis, Duncan’s multiple-range tests for all measured parameters are presented in Table 3.

Untreated specimens from all wood species exhibited baseline color attributes characterized by relatively high lightness (L*), moderate redness (a*), and yellowness (b*). Polystyrene treatment caused minimal discoloration, reflected by minor ΔE values ranging between 2.0 and 2.7, with an average of 2.3, indicating slight alteration from the original color. In contrast, specimens treated with CCA exhibited significantly higher discoloration, with ΔE values notably elevated across all species, ranging from 16.4 to 27.7, with an average of 22.2, resulting in a difference (Hrčková et al. [29]). This discoloration was associated with marked decreases (−5.5 point) in L* (darkening), substantial reductions (−9.3 point) in a* (shift towards green), and considerable increases (+18.0 point) in b* (intensified yellowness).

Among all tested species, *Falcataria moluccana* showed the greatest sensitivity to CCA treatment, with the highest ΔE of 27.7, followed by *Anisoptera thurifera* (24.5), *Octomeles sumatrana* (20.2), and *Anthocephalus cadamba* (16.4), suggesting pronounced susceptibility to chemical-induced color changes. These findings highlight the influence of preservative treatment type on wood coloration, which is essential for aesthetic considerations and consumer acceptance in wood product applications.

#### 3.1.2. Wood Density

Wood density is defined as the mass of wood per unit volume. Generally, in addition to being positively correlated with mechanical properties [34], higher wood density also tends to confer greater resistance to biodeterioration [35]. Wood density and moisture content of each wood species and treatment are presented in Table 4. Referring to the table, the wood used for this study ranged from 0.24 g/cm^3^ for *Falcataria moluccana* to 0.59 g/cm^3^ for *Anisoptera thurifera.* According to the Malaysian standard, all the woods are classified as light hardwood, which have a density below 0.72 g/cm^3^ [36].

Referring to the analysis of variance in Table 2, wood species and treatment significantly affect wood density. Wood species influence density due to inherent differences in characteristics such as cell wall thickness, lumen diameter, and other anatomical features. These characteristics are affected by the growth rate. As mentioned by Orwa et al. [37], *Falcataria* sp. exhibits a high growth rate, reaching a height of 25 m and a diameter at breast height (DBH) of 17 cm within just six years. Consequently, the wood density is very low. In contrast, *Anisoptera* sp. has a slower growth rate, with an average annual diameter increment of only 1.5 cm, resulting in a higher wood density compared to the other species used in this study.

The treatment also affects wood density, even though neither the polystyrene nor the preservative treatment chemically reacts with the wood components. Instead, the styrene monomer and preservative solution penetrate the wood’s void spaces. In the polystyrene treatment, the styrene monomer undergoes polymerization, initiated by a catalyst and heat, resulting in polystyrene filling the voids, as shown in Figure 4, which presents the Scanning Electron Microscopy (SEM) image of *Anisoptera thurifera* wood. In the preservation treatment, the solvent evaporates, leaving solid preservative material in the voids. Both treatments increase wood density through polymer loading (%) in the case of polystyrene impregnation and retention (kg/m^3^) in the preservation process. The polymer loading and retention values for each wood species are presented in Table 5.

The integration of SEM and EDS techniques provided confirmation of polystyrene (PS) infiltration within the *Anisoptera* sp. (A) wood matrix. The EDS elemental maps show clear differences in carbon and oxygen distribution between untreated wood A and polystyrene-impregnated wood A. In the untreated wood sample (A_Control_C), carbon is present throughout the structure, as expected due to the cellulose and lignin content, with intensity reaching about 19. In contrast, the polystyrene-impregnated wood (A_PS_C) displays a more uniform and dense carbon signal. Although the maximum scale value is slightly lower (around 16), the increased continuity indicates that carbon-rich polystyrene has infiltrated the wood and filled voids, resulting in a more consistent carbon distribution. For oxygen content, the untreated wood (A_Control_O) shows a strong and uniform signal, reflecting the natural abundance of cellulose [38], with counts up to about 13. In the impregnated wood (A_PS_O), the oxygen distribution appears less intense and more uneven. Despite a similar maximum scale (around 14), oxygen is visibly masked in areas of high carbon signal, suggesting that the polymer has partially covered or displaced the oxygen-rich surfaces.

These chemical patterns were corroborated by morphological evidence captured through SEM imaging. In untreated specimens, the cellular architecture, including vessels and fibers, appeared intact, with clean lumens and smooth wall surfaces characteristic of natural wood. In contrast, treated samples revealed a markedly different microstructure: numerous vessels were partially or completely filled, and cell wall surfaces appeared irregular or textured, indicating polymer deposition. Such alterations imply substantial penetration and structural anchoring of polystyrene, which are likely to contribute to reduced porosity and improved dimensional integrity [39,40].

The synergy of elemental and morphological data thus underscores the effectiveness of polystyrene impregnation in modifying wood properties. These findings align with earlier research demonstrating that polymer treatments can significantly enhance the durability and water resistance of otherwise low-performance wood species, through both chemical and structural reinforcement mechanisms [41,42].

Based on the data in Table 5, the higher polymer loading in polystyrene-treated wood is attributed to the high concentration of styrene monomer and the polymerization mechanism, which allows extensive filling of cell lumens and capillaries, particularly in low-density species with large void volumes. Conversely, the CCA treatment operates by chemical fixation of preservative salts to cell wall components, and the 3% concentration limits the total retention achievable within the same pressure and time conditions. This fundamental difference in mechanisms, physical polymer filling versus chemical fixation, results in significantly different amounts of deposited solids (Table 5). These findings are consistent with earlier studies demonstrating that monomer polymerization leads to higher mass gain and density increases compared to aqueous preservatives [43,44].

Regarding wood species, *Anisoptera* sp. had the lowest polymer loading because it had the highest density. As a result, the wood had fewer voids, allowing only a small amount of styrene monomer to penetrate, which led to lower polymer loading. On the other hand, *Octomeles* sp. and *Falcataria* sp. showed similarly high polymer loading, as both species have low densities and therefore contain more void space for the monomer to occupy.

In terms of retention, only *Octomeles* sp. met the Indonesian standard minimum of 8 kg/m^3^. To increase retention, the concentration of the preservative solution could be raised, the immersion time extended, or a higher pressure applied during the treatment process.

The increase in wood density corresponds to the polymer loading and retention values for each wood species, as shown in Table 4, which presents the density and moisture content. Furthermore, to analyze the effect of wood species and treatment on density and moisture content, Duncan’s multiple range test was applied, and the results are presented in Table 6.

Referring to Table 6, all the wood samples are still categorized as light hardwoods, and the order of wood densities remains the same as in the untreated conditions. This indicates that the density of each wood species is significantly different from the others.

In terms of treatment, wood treated with polystyrene showed the highest density, which corresponds to the high polymer loading. The increase in density from untreated wood (0.42 g/cm^3^) to polystyrene-treated wood (0.64 g/cm^3^) is consistent with the polymer loading, which reached 145% in this study. It can be concluded that higher polymer loading in polystyrene treatment, as well as higher retention of the CCA preservative, results in a greater increase in wood density. This increase is expected to enhance the wood’s resistance to biodeterioration.

#### 3.1.3. Wood Moisture Content

Wood is a hygroscopic material because it contains hydroxyl groups that have an affinity for water and water vapor. The amount of water absorbed by wood is expressed as its moisture content. In Bogor, a tropical area, the equilibrium moisture content ranges between 12% and 18% [45]. Based on the results, the moisture content of all wood samples falls within this equilibrium range for the region.

The moisture content of each wood species and treatment is presented in Table 4, while the summary of the variance analysis is shown in Table 2. According to Table 2, wood moisture content is significantly affected by both wood species and treatment. Variations in wood density can influence moisture content: lower-density woods typically have thinner cell walls, more voids, and lower cellulose content, which increases their accessibility to water. As a result, untreated *Falcataria* sp. and *Octomeles* sp. exhibit higher moisture content compared to the other two species, as water can more easily interact with the cellulose through hydrogen bonding throughout most parts of the wood, a phenomenon less pronounced in higher-density woods.

In terms of treatment, polystyrene-treated wood had the lowest moisture content (9.4%) because polystyrene is hydrophobic and reduces the wood’s surface area available for water interaction, especially by coating the lumen areas. As a result, while the weight of the wood increases, its ability to absorb water decreases, leading to lower moisture content. Compared to untreated wood, which has a moisture content of 15.8%, the moisture content of polystyrene-treated wood is reduced by 40.5%.

CCA-treated wood undergoes an air-drying process during treatment, resulting in a final moisture content of 13.4%. The air-drying stage plays a significant role in improving the penetration efficiency of the CCA preservative. Lowering the initial moisture content of the wood increases its permeability, allowing the preservatives to diffuse more effectively into the wood fibers [46].

Additionally, during the treatment process, CCA chemicals chemically bind with wood components such as lignin and cellulose, creating stable, water-insoluble complexes. These complexes reduce the wood’s capacity to retain additional moisture, contributing to the lower final moisture content after treatment [47]. Air-drying also facilitates the evaporation of excess water introduced during the pressure treatment. By removing this additional moisture, treated wood attains a stable moisture content, reducing susceptibility to future swelling and shrinkage, thereby improving dimensional stability [48].

However, moisture content plays a nuanced role in preservative fixation and leachability. According to Kaldas and Cooper [49], lower moisture content can slow the chromium fixation process, potentially increasing the leaching of CCA components. Similarly, studies by Kim and Ra [50] have suggested that wood conditioned under drying conditions experiences slower preservative fixation compared to wood treated under non-drying conditions. In contrast, Lee [47] reported that CCA-treated lumber, when conditioned under the same relative humidity and temperature as untreated lumber, exhibited a higher equilibrium moisture content (EMC), which highlights the complexity of post-treatment moisture dynamics.

Ultimately, the decision to air-dry CCA-treated wood during treatment represents a balance between enhancing preservative penetration, ensuring stable chemical fixation, and controlling moisture content. In this study, the final moisture content of CCA-treated wood was decreased by 15% compared to untreated wood, contributing to improved performance and protection characteristics of the final product.

#### 3.1.4. Wood Weight Loss

In tropical areas, wood deterioration caused by termite attacks is a serious problem [51], as buildings constructed with wood materials are highly susceptible to termite damage. The degree of termite attack can be measured by the weight loss of the wood. In this study, the wood was exposed in Papua New Guinea for four months. The weight loss and protection level of each wood and treatment are presented in Table 7, while the attacked wood samples are shown in Figure 5.

From the table, it can be observed that the weight loss of untreated wood has the highest value (18.92%), indicating that the wood is seriously attacked. Referring to the variance analysis summary in Table 2, both factors of wood species and treatment significantly affected wood weight loss. For further data analysis, the results of Duncan’s multiple-range test are presented in Table 8.

The four untreated wood species fall into durability class 5, the lowest durability class [52]. As shown in Table 8, *Falcataria* sp. exhibited the highest weight loss, which was significantly different from the other three species, while the differences among the remaining three species were not statistically significant. *Falcataria* sp., having the lowest density among the four species, also showed the greatest weight loss in the untreated condition, making it highly susceptible to subterranean termite attacks. For this reason, *Falcataria* sp. is strongly recommended for inclusion in termite resistance testing [53].

In terms of treatment, CCA-preserved wood proved to be highly effective in preventing termite attacks, with a weight loss of only 1.87%, compared to 18.9% in untreated wood. However, this chemical compound is toxic not only to termites but also to other living organisms, including humans. As a result, the use of CCA preservatives has been banned in many countries.

Polystyrene-treated wood is reasonably effective in increasing durability, as indicated by a weight loss of only 6.27%, or about one-third of that of untreated wood. This result is consistent with previous studies by Acosta et al. [43] and Hadi et al. [17,33]. The treated wood has a higher density than untreated wood, and since polystyrene is a type of plastic that termites do not consume, it is also considered safe for humans. Given its safety for living organisms, polystyrene-treated wood could be considered for future wood products. Moreover, it offers improved physical and mechanical properties compared to untreated wood [18]. Although polystyrene treatment has offered several advantages, certain limitations should be taken into account, including the need for proper protection during the handling of styrene monomer in the wood modification process and the assessment of the environmental fate of polystyrene in wood.

Based on field observations, termite attacks on the wood samples were evident, as shown in Figure 6. According to the identification procedures and with reference to Thistleton et al. [54], the termite species was identified as *Mastotermes darwiniensis*.

### 3.2. Wood Protection Level

The protection levels obtained from testing four wood species, namely *Anisoptera thurifera*, *Octomeles sumatrana*, *Falcataria moluccana*, and *Anthocephalus cadamba*, under untreated, polystyrene, and CCA-treated conditions, are presented in Table 7. The untreated samples of all four tropical species showed lower protection levels ranging from 5.6 to 7.5, reflecting significant vulnerability to environmental factors such as moisture intrusion and insect infestation, especially termite attack. *Anthocephalus cadamba* exhibited the lowest protection level among untreated samples (5.6 ± 1.2), indicating greater susceptibility to degradation compared to other species.

In contrast, the polystyrene-impregnated and CCA-treated samples showed significantly improved protection levels across all wood species, with values consistently reaching near maximum (9.8 to 10.0). *Octomeles sumatrana* notably achieved a perfect protection rating (10.0 ± 0.0) for both polystyrene and CCA treatments, highlighting exceptional efficacy and complete protection against environmental stressors. The observed high protection levels from polystyrene treatment are attributed to effective encapsulation, reducing moisture ingress, and providing an effective physical barrier against termites. Similarly, the efficacy of CCA treatment is due to chemical interactions of chromium, copper, and arsenate compounds within the wood structure, offering robust resistance against biodeterioration.

Both treatments demonstrated comparable effectiveness, suggesting that polystyrene impregnation could serve as a promising alternative to conventional chemical preservative treatments like CCA. Polystyrene offers additional advantages in terms of environmental safety by minimizing the risks of chemical leaching and environmental contamination commonly associated with traditional preservatives such as CCA. Although polystyrene-impregnated wood requires higher polymer loading, the non-polar nature of the polymer allows it to remain stable within the wood’s cavities or lumen cells, reducing the likelihood of leaching in water or under naturally wet conditions. Furthermore, polystyrene is relatively stable and considered safe for humans, animals, and the environment under normal environmental conditions. In contrast, CCA not only tends to leach in moist environments but also contains biocides that are regarded as harmful and toxic to living organisms and the ecosystem. These findings highlight the potential of polystyrene impregnation as a sustainable wood preservation method, offering significant durability improvements while addressing environmental and health concerns linked to chemical-based treatments.

## 4. Conclusions

Based on the discussion above, it can be concluded that polystyrene-treated wood exhibits only minor discoloration, with color changes visible primarily under high-quality filters. In contrast, CCA-treated wood shows noticeable color changes compared to untreated wood, with a dominant shift toward yellow, followed by green hues and slight darkening. The combined SEM and EDS results clearly demonstrated that polystyrene successfully penetrated and integrated into the wood structure, both chemically and physically. This treatment enhanced the wood’s properties by filling voids, reducing porosity, and potentially increasing durability and water resistance.

All four wood species can be successfully impregnated with polystyrene, with an average polymer loading of 147%. *Octomeles* sp. achieved the highest polymer loading (234%), while *Anisoptera* sp. *Anisoptera* sp. had the lowest (12%). CCA preservation resulted in an average retention of 8.4 kg/m^3^, with *Octomeles* sp. again showing the highest value (13.5 kg/m^3^) and *Falcataria* sp. the lowest (5.7 kg/m^3^). The moisture content of all wood samples was consistent with ambient conditions in Lae, Papua New Guinea. Polystyrene-treated wood had the lowest moisture content (9.4%), followed by CCA-treated wood (13.4%) and untreated wood (15.8%). According to Indonesian Standard SNI 7207-2014 [55], all four tested wood species fall into durability class 5, indicating the lowest durability category. CCA-preserved wood proved highly effective in preventing termite attacks across all species; however, it poses health and environmental risks due to its toxic nature. In contrast, polystyrene-treated wood also demonstrated strong resistance to subterranean termite attacks, with lower weight loss and higher protection levels than untreated wood. Regarding polystyrene itself, we acknowledge that it is not biodegradable. However, unlike CCA-treated wood, polystyrene does not release toxic metals during use, making it a promising alternative for future wood protection applications.

Based on the above-mentioned advantages and limitations of polystyrene treatment, future studies are recommended to investigate the long-term behavior of polystyrene-treated wood under solar exposure and field conditions, investigate detailed molecular analysis to better understand how the properties of the in situ polymer relate to durability and other performance outcomes, assess the environmental fate of polystyrene in wood, and explore the potential of using biodegradable or bio-based monomers for more sustainable wood protection technologies.

## Figures and Tables

**Figure 1 polymers-17-01945-f001:**
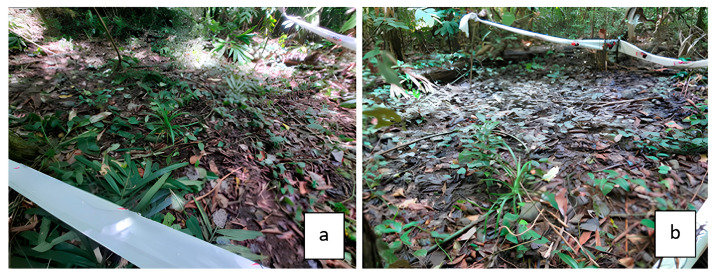
Field test conditions of wood specimens before the test (**a**) and after the test (**b**).

**Figure 2 polymers-17-01945-f002:**
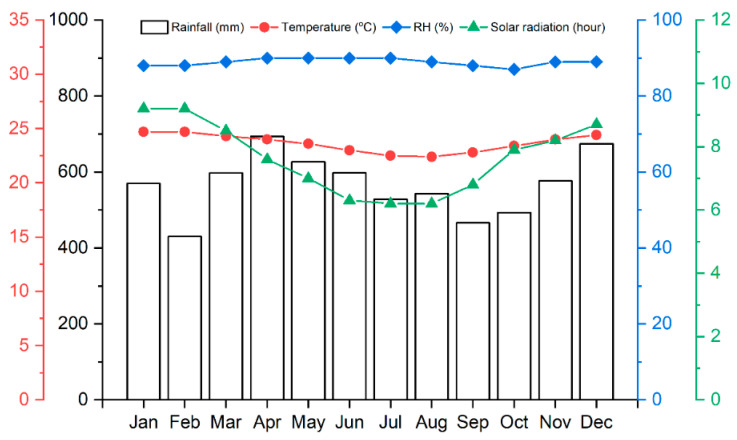
Ombrophobic diagram in Lae, Papua New Guinea. Source [30].

**Figure 3 polymers-17-01945-f003:**
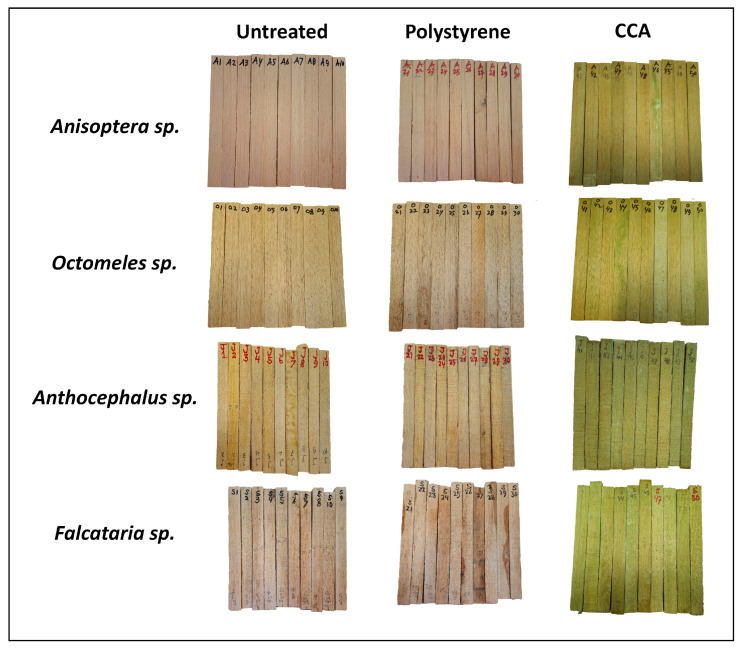
Color of untreated, polystyrene-impregnated, and CCA-preserved woods before field exposure.

**Figure 4 polymers-17-01945-f004:**
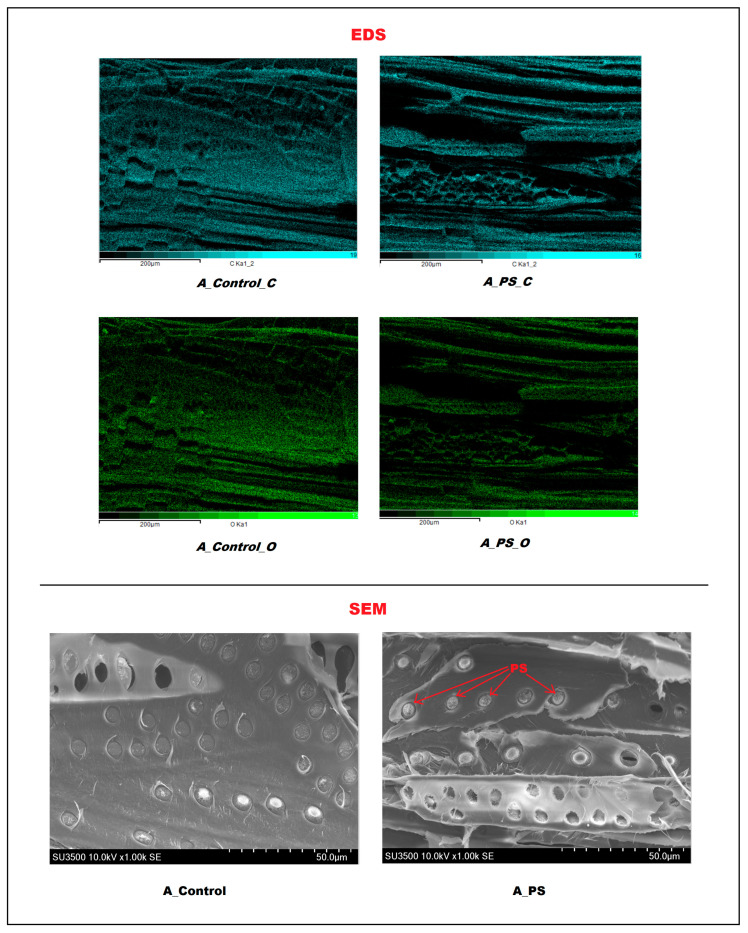
EDS and SEM images of untreated *Anisoptera thurifera* (A_control) and polystyrene-impregnated *Anisoptera thurifera* (A_PS) (polystyrene deposits are indicated by the arrows).

**Figure 5 polymers-17-01945-f005:**
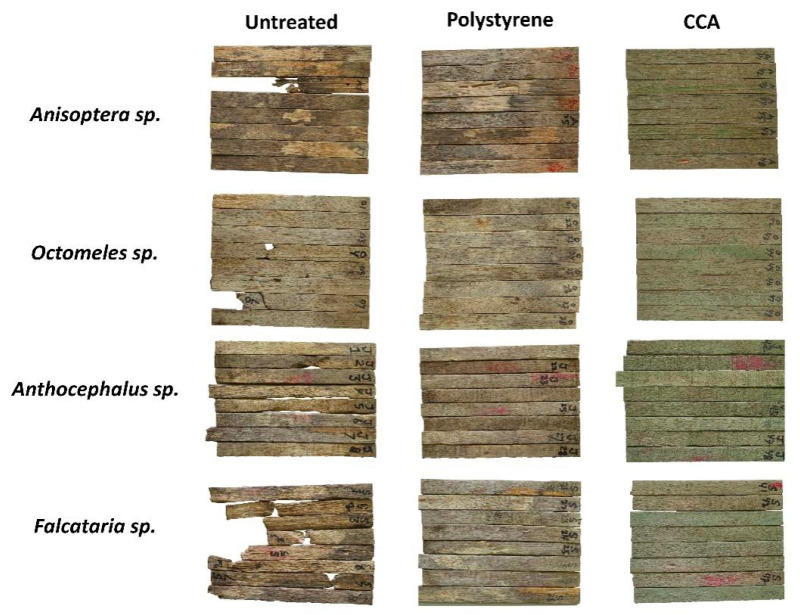
Attacked wood after exposure in the field.

**Figure 6 polymers-17-01945-f006:**
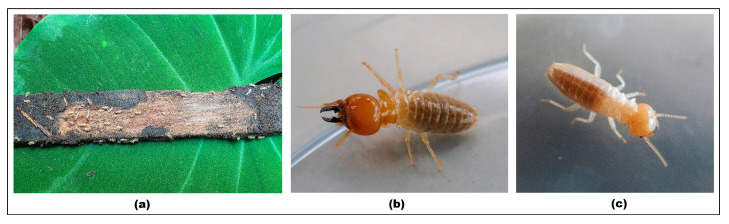
Attacked wood (**a**), soldier termite (**b**), and worker termite (**c**).

**Table 1 polymers-17-01945-t001:** Color and color change indexes of wood samples.

Wood Species	Treatment	L*	a*	b*	ΔE
*Anisoptera thurifera*	Untreated	63.8 (3.5)	5.0 (1.2)	15.8 (0.7)	
	Polystyrene	64.1 (4.4)	4.7 (1.9)	16.6 (1.5)	2.7 (1.2)
	CCA wood	57.8 (4.5)	−3.0 (0.9)	37.4 (3.6)	24.5 (2.7)
*Octomeles sumatrana*	Untreated	68.1 (1.7)	1.8 (0.6)	23.6 (2.8)	-
	Polystyrene	68.7 (2.0)	1.9 (0.9)	22.2 (3.0)	2.0 (0.8)
	CCA wood	63.9 (3.6)	−5.1 (1.0)	41.5 (4.1)	20.2 (4.0)
*Falcataria moluccana*	Untreated	67.2 (1.5)	4.5 (1.2)	17.2 (2.6)	-
	Polystyrene	67.9 (2.0)	5.1 (0.9)	17.4 (2.8)	2.2 (1.4)
	CCA wood	64.0 (2.9)	−8.3 (2.1)	41.2 (2.9)	27.7 (3.0)
*Anthocaphalus cadamba*	Untreated	67.1 (2.4)	2.3 (1.5)	26.6 (1.6)	-
	Polystyrene	66.9 (2.8)	2.4 (1.2)	26.1 (0.8)	2.3 (1.0)
	CCA wood	58.0 (4.4)	−7.1 (0.6)	34.8 (2.3)	16.4 (1.6)

Note: values in parentheses are standard deviations.

**Table 2 polymers-17-01945-t002:** Variance analysis of wood exposed in the field.

Parameter	Wood Species	Treatment
L* (lightness)	**	**
a* (red-green)	**	**
b* (yellow-blue)	**	**
∆E (discoloration)	**	**
Density	**	**
Moisture content	**	**
Weight loss	**	**
Protection level	**	**

Note: ** indicates highly significant differences at *p* ≤ 0.01.

**Table 3 polymers-17-01945-t003:** Duncan’s multiple-range test of wood color.

Parameter	Wood Species				Treatment		
	*Anisoptera * *thurifera*	*Octomeles* *sumatrana*	*Falcataria* *moluccana*	*Anthocephalus* *cadamba*	Untreated	Polystyrene	CCA Wood
L* value	66.3 c	66.9 c	61.9 a	64.1 b	66.5 e	66.9 e	61.0 d
a* value	0.5 b	−0.4 a	2.2 c	−0.8 a	3.5 e	3.6 e	−5.9 d
b* value	25.2 b	29.1 c	23.2 a	29.2 c	20.8 d	20.6 d	38.7 e
∆E treatment	13.6 b	11.1 a	14.8 b	9.3 a	-	2.3 c	22.2 d

Note: the same letter following the value of wood species and treatment in a row indicates no significant difference.

**Table 4 polymers-17-01945-t004:** Wood density and moisture content of the wood sample.

Wood Species	Treatment	Density (g/cm^3^)	Moisture Content (%)
*Anisoptera thurifera*	Untreated	0.59 (0.01)	14.29 (0.29)
	Polystyrene	0.62 (0.02)	9.25 (0.48)
	CCA wood	0.61 (0.01)	12.33 (0.67)
*Octomeles sumatrana*	Untreated	0.27 (0.02)	17.03 (0.59)
	Polystyrene	0.68 (0.02)	9.94 (0.40)
	CCA wood	0.32 (0.02)	11.03 (0.42)
*Anthocephalus cadamba*	Untreated	0.48 (0.06)	15.04 (0.18)
	Polystyrene	0.68 (0.02)	9.25 (0.12)
	CCA wood	0.51 (0.04)	15.03 (0.89)
*Falcataria moluccana*	Untreated	0.24 (0.01)	17.03 (0.59)
	Polystyrene	0.54 (0.07)	9.23 (0.13)
	CCA wood	0.27 (0/03)	15.06 (0.62)

Note: values in parentheses are standard deviations.

**Table 5 polymers-17-01945-t005:** Polymer loading and retention of each process for each wood species.

Treatment	Unit	*Anisosptera* sp.	*Octomeles* sp.	*Anthocephalus* sp.	*Falcataria* sp.
Polystyrene	Polymer loading, %	12.3 (1.5)	233.6 (25.5)	132.5 (15.8)	210.8 (43.0)
Preservation	Retention, kg/m^3^	7.4 (0.9)	13.5 (0.8)	7.1 (0.3)	5.7 (0.7)

Note: values in parentheses are standard deviations.

**Table 6 polymers-17-01945-t006:** Duncan’s multiple-range test of density and moisture content.

Response	Wood Species				Treatment		
	*Anisoptera* sp.	*Octomeles* sp.	*Anthocephalus* sp.	*Falcataria* sp.	Untreated	Polystyrene	Preserved
Density	0.68 d	0.42 b	0.55 c	0.35 a	0.42 e	0.64 g	0.45 f
MC	12.0 a	12.7 b	13.1 bc	13.8 c	15.8 c	9.4 a	13.4 b

Note: The same letter following the value of wood species and treatment in a row indicates no significant difference.

**Table 7 polymers-17-01945-t007:** Weight loss and protection level of the wood sample.

Wood Species	Treatment	Weight Loss (%)	Protection Level
*Anisoptera thurifera*	Untreated	14.73 (24.70)	7.9 (3.2)
	Polystyrene	6.49 (2.47)	9.8 (0.5)
	CCA wood	0.56 (0.46)	9.8 (0.5)
*Octomeles sumatrana*	Untreated	14.89 (9.82)	7.3 (2.2)
	Polystyrene	2.20 (0.51)	10.0 (0.0)
	CCA wood	0.72 (0.41)	10.0 (0.0)
*Anthocephalus cadamba*	Untreated	18.53 (11.99)	5.6 (1.7)
	Polystyrene	7.36 (2.29)	9.8 (0.5)
	CCA wood	2.28 (2.23)	9.8 (0.5)
*Falcataria moluccana*	Untreated	30.31 (13.28)	7.5 (2.3)
	Polystyrene	9.04 (6.00)	9.8 (0.5)
	CCA wood	3.91 (2.74)	9.8 (0.5)

Note: values in parentheses are standard deviations.

**Table 8 polymers-17-01945-t008:** Duncan’s multiple-range test of weight loss and protection level.

Response	Wood Species				Treatment		
	*Anisoptera* sp.	*Octomeles* sp.	*Anthocephalus* sp.	*Falcataria* sp.	Untreated	Polystyrene	CCA
Weight loss	6.34 a	5.93 a	9.39 a	14.42 b	18.92 c	6.27 d	1.87 e
Protection	8.58 b	8.45 b	8.08 ab	7.67 a	7.06 c	8.15 d	9.37 e

Note: the same letter following the value of wood species and treatment in a row indicates no significant difference.

## Data Availability

The original contributions presented in this study are included in the article. Further inquiries can be directed to the corresponding authors.

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
