# Peer review of "Performance of Polystyrene-Impregnated and CCA-Preserved Tropical Woods Against Subterranean Termites in PNG Field and Treatment-Induced Color Change"

_polymers, 2025, doi:10.3390/polym17141945_

Round 1
Reviewer 1 Report
Comments and Suggestions for Authors
Impregnation of wood with styrene monomers is a chemical modification technique that has been extensively studied (book from D. O. Stolf et al. for example). The authors must presented this results in introduction part.
Why the authors focused the title on wood color? they just determined the color after different treatments. A study of discolouration by solar irradiation could have been interesting.
It is well known that wood treated by CCA is rather greenish.
They is no information concerning the polystyrene obtained after styrene polymerization. What are the molar masses, the dispersity.
The authors wrote that "is CCA treatment living organisms, including humans". This is right by they must discussed the toxicity of styrene used for the treatment and the polystyrene is not biodegradable.
Author Response
Comments and Suggestions from Reviewer 1
Impregnation of wood with styrene monomers is a chemical modification technique that has been extensively studied (book from D. O. Stolf et al. for example). The authors must presented this results in the introduction part.
Answer:
Thank you for the comment. We have added in the text regarding this concern in the introduction.
«As alternative solutions to these growing environmental concerns, wood modifications including thermal treatments, impregnation, and chemical modifications have been increasingly studied and industrially implemented [8-16]. In particular, wood chemical modification such as furfurylation or acetylation are considered capable of improving wood properties, specifically its dimensional stability and durability against biodeterioration, including its resistance against termite attacks. However, these modifications generally require a higher investment, especially in the quantity of the reactive chemicals used and the recovery of the post-impregnating solutions [14].
Based on the above limitation, in the present study, the impregnation of wood using styrene monomer can be another alternative to improve wood properties, particularly its durability against biodeterioration. The recovery issue of the post-impregnating solution can be eliminated since the styrene monomer is relatively stable and can be reused many times. Polymer impregnation technology, particularly using polystyrene, has emerged as a promising approach to improve the durability and resistance of wood against biological attacks [17]. Polystyrene impregnation can significantly reduce moisture uptake, enhance dimensional stability, and decrease susceptibility to biodegradation agents [18]. Despite these advantages, limited research has focused specifically on the application and performance of polystyrene impregnation for tropical wood species, especially under actual field conditions where termite pressure is high [19,20]. Therefore, to further investigate the termite resistance of polystyrene-modified wood under real field conditions, a series of treatments, including impregnation with a well-known wood preservative, should be conducted and evaluated [21]. In addition, although polystyrene-modified wood is not a new technology, the application of this technique to certain wood species from Papua New Guinea (PNG), which are relatively new in the field of wood modification, offers valuable insights for readers, particularly for PNG citizens who wish to enhance the durability and stability of their native wood resources.»
- Why the authors focused the title on wood color? they just determined the color after different treatments. A study of discolouration by solar irradiation could have been interesting.It is well known that wood treated by CCA is rather greenish.
Answer:
Thank you for the comment. In this study, we focused on color in the title because the visual change in treated wood can be a practical and immediate indicator of chemical modification, which many practitioners consider important. However, we recognize that our study only examined the initial color changes after treatment and did not extend to discoloration over time due to solar exposure or field test effect. In response to your comment, we have adjusted the title to better reflect the actual scope of the work, highlighting treatment effects and resistance rather than general discoloration. We’ve also added a note in the revised manuscript acknowledging that examining long-term color stability under sunlight would be a valuable area for further research.
The new title of the manuscript:
"Performance of Polystyrene-Impregnated and CCA-Preserved Tropical Woods Against Subterranean Termites in PNG Field and Treatment-Induced Color Change"
- Thereis no information concerning the polystyrene obtained after styrene polymerization. What are the molar masses, the dispersity.
Answer:
Thank you for the comment. Our study was primarily focused on assessing how the in situ polymerization affected the wood’s properties, specifically, its appearance and biological resistance, rather than on characterizing the polymer itself in detail. Because the polymerization took place directly inside the wood without solvents, we did not extract the polystyrene for molar mass or dispersity measurements.
That said, we agree that this information would add valuable insight into the process and could help link the polymer’s characteristics to the performance of treated wood. We have now clarified this limitation in the manuscript and suggested that future investigations should include detailed molecular analysis to better understand how the properties of the in situ polymer relate to durability and other performance outcomes.
In the conclusion we have added this limitation for future work:
«Based on the above-mentioned advantages and limitations of polystyrene treatment, future studies are recommended to investigate the long-term behavior of polystyrene-treated wood under solar exposure and field conditions, investigate detailed molecular analysis to better understand how the properties of the in situ polymer relate to durability and other performance outcomes, assess the environmental fate of polystyrene in wood, and explore the potential of using biodegradable or bio-based monomers for more sustainable wood protection technologies»
- The authors wrote that "CCA is hazardous to living organisms, including humans".This is right, but they must discussed the toxicity of styrene used for the treatment and the polystyrene is not biodegradable.
Answer:
Thank you for highlighting this important aspect. You are absolutely right that the risks associated with styrene should be addressed alongside the hazards of CCA. We have added a discussion in the revised manuscript noting that:
«Although polystyrene treatment has offered several advantages, certain limitations should be taken into account, including the need for proper protection during the handling of styrene monomer in the wood modification process and the assessment of the environmental fate of polystyrene in wood.»
Regarding polystyrene, we acknowledge that it is not biodegradable, which raises environmental concerns. However, unlike CCA-treated wood, polystyrene does not release toxic metals during use. We have also suggested that future work explore alternative monomers or biodegradable polymers to address these sustainability challenges.
In the conclusion part we have added some sentences regarding this issue:
«Regarding polystyrene itself, we acknowledge that it is not biodegradable. However, unlike CCA-treated wood, polystyrene does not release toxic metals during use, making it a promising alternative for future wood protection applications.
Based on the above-mentioned advantages and limitations of polystyrene treatment, future studies are recommended to investigate the long-term behavior of polystyrene-treated wood under solar exposure and field conditions, assess the environmental fate of polystyrene in wood, and explore the potential of using biodegradable or bio-based monomers for more sustainable wood protection technologies.»
«

Reviewer 2 Report
Comments and Suggestions for Authors
Jingjing Liao et al., in their manuscript entitled "Discoloration and Resistance of Polystyrene Impregnated and Preserved Four Tropical Woods Exposed to Subterranean Termite in Papua New Guinea Field," tried to determine the discoloration and resistance of polystyrene-impregnated and CCA-preserved woods. They found that the color change in polystyrene-impregnated wood was small, whereas CCA-preserved wood exhibited a different color compared to untreated wood. However, some of the limitations listed below were observed:
- The manuscript contains typos and spelling mistakes. They should be corrected.
- The novelty of this work is absent. The authors should illustrate it in more detail in both the abstract and the introduction.
- The introduction is weak. The authors should add more information about other current methods for enhancing biodeterioration attacks of the tropical wood species and their effect on changes in wood color, and their cons and pros.
- The properties of used wood species in this work are absent. The authors should add more information about these properties.
- The authors wrote, “The weight percentage gain (WPG) of each wood sample was measured as previously described”. The authors should shortly describe the weight percentage gain.
- The authors wrote, “Wood density could refer to its mechanical properties, a higher density could have higher mechanical properties”. It is better to provide the mechanical properties for better investigation.
- The resolution of Figure 4 should be improved.
- The authors wrote, “Based on the data in Table 5, the polystyrene treatment resulted in high polymer loading because the styrene monomer used had a high concentration, as it was purchased directly from a chemical supplier. In contrast, the CCA solution had a concentration of only 3%. Due to this difference, the amount of solid chemical deposited in the wood voids varied significantly between the two treatments.” This is considered a very simple explanation without any scientific evaluation.
- The authors wrote, “Both treatments demonstrated comparable effectiveness, suggesting that polystyrene impregnation could serve as a promising alternative to conventional chemical preservative treatments like CCA. Polystyrene offers additional advantages in terms of environmental safety by minimizing the risks of chemical leaching and environmental contamination commonly associated with traditional preservatives such as CCA.”. The main question here is why is polystyrene impregnation considered better? The discussion provided is regarded as general.
- The authors should add all DOI links for the cited references.
The manuscript contains typos and spelling mistakes.
Author Response
Comments and Suggestions for Authors (2nd Reviewer)
Jingjing Liao et al., in their manuscript entitled "Discoloration and Resistance of Polystyrene Impregnated and Preserved Four Tropical Woods Exposed to Subterranean Termite in Papua New Guinea Field," tried to determine the discoloration and resistance of polystyrene-impregnated and CCA-preserved woods. They found that the color change in polystyrene-impregnated wood was small, whereas CCA-preserved wood exhibited a different color compared to untreated wood. However, some of the limitations listed below were observed:
- The manuscript contains typos and spelling mistakes. They should be corrected.
Answer:
Thank you for the comment. Some typos and spelling mistakes have been corrected.
- The novelty of this work is absent. The authors should illustrate it in more detail in both the abstract and the introduction.
Answer:
Thank you for the comment. In PNG, almost all wood species are treated with biocides such as CCA (Chromated Copper Arsenate) or ACQ (Alkaline Copper Quaternary) on an industrial scale to preserve wood. However, as environmental concerns have grown in many aspects, the use of biocides such as CCA and ACQ for wood preservation has become limited or even restricted in many countries, including PNG in the near future. As an alternative to replace these biocides, wood modification treatment has become increasingly interesting to researchers and industry. Some environmentally friendly chemicals have been implemented, such as furfurylation and acetylation, and polystyrene impregnation can also be considered an alternative to environmentally friendly wood modification methods.
Although polystyrene-modified wood is not a new technology, the application of this technique to certain wood species from Papua New Guinea (PNG), which are relatively new in the field of wood modification, offers valuable insights for readers, particularly for PNG citizens who wish to enhance the durability and stability of their native wood resources.
Moreover, polystyrene impregnation presents several advantages over other common wood modification methods. The impregnating solution can be reused for subsequent treatments, and the curing process occurs at a relatively low temperature (103 °C). Due to the relatively low reactivity of styrene monomers with wood components, this treatment helps preserve the mechanical properties of the wood, and in some cases, can even improve them.
- The introduction is weak. The authors should add more information about other current methods for enhancing biodeterioration attacks of the tropical wood species and their effect on changes in wood color, and their cons and pros.
Answer:
Thank you for the comment. We have added additional information on current methods used to enhance resistance against biodeterioration in tropical wood species, including their advantages and disadvantages, as well as their effects on wood color. These revisions have been incorporated into the introduction section of the manuscript.
- The properties of used wood species in this work are absent. The authors should add more information about these properties.
Answer:
Thank you for the comment. We have added some additional properties of the used wood species in the text with their supported references.
«Two wood species from Papua New Guinea, namely Anisoptera thurifera (density: 0.6 - 0.65 g/cm3) [Richter & Dallwitz 2000a], and Octomeles sumatrana (density: 0.25 - 0.38 g/cm3) [Richter & Dallwitz 2000b], were obtained from a timber shop in East Taraka, Lae, Papua New Guinea. The other two wood species, Anthocephalus cadamba (density: 0.29 - 0.56 g/cm3 [Orwa et al. 2009] and Falcataria moluccana (density: 0.23 - 0.50 g/cm3) [Richter & Dallwitz 2000c]»
- The authors wrote, “The weight percentage gain (WPG) of each wood sample was measured as previously described”. The authors should shortly describe the weight percentage gain.
Answer:
Thank you for the comment. The weight percentage gain (WPG) was calculated as the percentage increase in oven-dry weight of the wood after treatment, relative to the oven-dry weight before treatment, using the formula:
WPG (%) = [(W1 − W0) / W0] × 100,
where W0 is the initial oven-dry weight and W1 is the oven-dry weight after impregnation and curing.
- The authors wrote, “Wood density could refer to its mechanical properties, a higher density could have higher mechanical properties”. It is better to provide the mechanical properties for better investigation.
Answer:
Thank you for the comment. Wood density could refer to its mechanical properties, a higher density could have higher mechanical properties.
Corrected to:
Wood density is defined as the mass of wood per unit volume. Generally, in addition to being positively correlated with mechanical properties [Bardet et al., 2003], higher wood density also tends to confer greater resistance to biodeterioration [24].
Ref:
Bardet, S., Beauchêne, J., & Thibaut, B. (2003). Influence of basic density and temperature on mechanical properties perpendicular to grain of ten wood tropical species. Annals of forest science, 60(1): 49-59. DOI: 10.1051/forest: 2002073
- The resolution of Figure 4 should be improved.
Answer:
Thank you for the comment. The resolution of Figure 4 has been improved, as well as its interpretation in the text.
- The authors wrote, “Based on the data in Table 5, the polystyrene treatment resulted in high polymer loading because the styrene monomer used had a high concentration, as it was purchased directly from a chemical supplier. In contrast, the CCA solution had a concentration of only 3%. Due to this difference, the amount of solid chemical deposited in the wood voids varied significantly between the two treatments.” This is considered a very simple explanation without any scientific evaluation.
Answer:
Thank you for the comment. The higher polymer loading in polystyrene-treated wood is attributed to the high concentration of styrene monomer and the polymerization mechanism, which allows extensive filling of cell lumens and capillaries, particularly in low-density species with large void volumes. Conversely, the CCA treatment operates by chemical fixation of preservative salts to cell wall components, and the 3% concentration limits the total retention achievable within the same pressure and time conditions. This fundamental difference in mechanisms, physical polymer filling versus chemical fixation, results in significantly different amounts of deposited solids (Table 5). These findings are consistent with earlier studies demonstrating that monomer polymerization leads to higher mass gain and density increases compared to aqueous preservatives (Acosta et al. 2021; Araújo et al. 2004).
Ref:
Acosta, A. P., de Avila Delucis, R., Amico, S. C., & Gatto, D. A. (2021). Fast-growing pine wood modified by a two-step treatment based on heating and in situ polymerization of polystyrene. Construction and Building Materials, 302, 124422. https://doi.org/10.1016/j.conbuildmat.2021.124422
Araujo, P. H. H. D., Sayer, C., Giudici, R., & Poco, J. G. (2002). Techniques for reducing residual monomer content in polymers: a review. Polymer Engineering & Science 42(7):1442-1468. https://doi.org/10.1002/pen.11043
- The authors wrote, “Both treatments demonstrated comparable effectiveness, suggesting that polystyrene impregnation could serve as a promising alternative to conventional chemical preservative treatments like CCA. Polystyrene offers additional advantages in terms of environmental safety by minimizing the risks of chemical leaching and environmental contamination commonly associated with traditional preservatives such as CCA.”. The main question here is why is polystyrene impregnation considered better? The discussion provided is regarded as general.
Answer:
Thank you for the comment. We have added some comparative information to the above statement.
Both treatments demonstrated comparable effectiveness, suggesting that polystyrene impregnation could serve as a promising alternative to conventional chemical preservative treatments like CCA. Polystyrene offers additional advantages in terms of environmental safety by minimizing the risks of chemical leaching and environmental contamination commonly associated with traditional preservatives such as CCA. Although polystyrene-impregnated wood requires higher polymer loading, the non-polar nature of the polymer allows it to remain stable within the wood’s cavities or lumen cells, reducing the likelihood of leaching in water or under naturally wet conditions. Furthermore, polystyrene is relatively stable and considered safe for humans, animals, and the environment under normal environmental conditions. In contrast, CCA not only tends to leach in moist environments but also contains biocides that are regarded as harmful and toxic to living organisms and the ecosystem. These findings highlight the potential of polystyrene impregnation as a sustainable wood preservation method, offering significant durability improvements while addressing environmental and health concerns linked to chemical-based treatments.
- The authors should add all DOI links for the cited references.
Answer:
Thank you for the comment. DOI links will be provided with the journal.
Comments on the Quality of English Language
The manuscript contains typos and spelling mistakes.
Submission Date
14 June 2025
Date of this review
25 Jun 2025 13:46:27

Reviewer 3 Report
Comments and Suggestions for Authors
The overall research is good, but there are some parts need more addressing, please consider the following parts.
- Did you monitor drying conditions (e.g., temperature, relative humidity), or just ambient air-drying?
- What purity grades were used for styrene monomer and potassium peroxydisulfate?
- Why was 600 mmHg vacuum chosen? Was this based on preliminary trials or a specific standard?
- Was 2% w/w determined experimentally, or adopted from prior literature?
- Did you control humidity during the 24 h polymerization step?
- Which termite species were predominant at the field site?
- Were L*, a*, and b* values measured immediately after treatment or after conditioning?
- Did you also calculate ΔE for all treatments? You could also mention whether the observed ΔE exceeds thresholds for perceptible or unacceptable discoloration.
- You may wish to briefly explain why increased density can be beneficial (e.g., improved mechanical performance, durability).
- Besides qualitative mapping, did you quantify the carbon content increase (e.g., atomic percent) to estimate polymer loading more precisely?
- Do you have cross-sectional images showing how deeply the polystyrene penetrated into the wood matrix?
- How did you control for potential artifacts (e.g., beam damage, outgassing of monomer residues) during imaging?
- How do you expect the observed reduction in porosity to affect moisture uptake, swelling, or dimensional stability during service?
- Is there a similar minimum standard for polystyrene-treated wood retention or loading in Indonesia or other countries?
- How do the retention levels observed in CCA-treated samples compare to performance benchmarks for decay or termite resistance?
- Was moisture content controlled before treatment, and did it influence polymer uptake?
- Was the reduction in moisture content for polystyrene-treated samples consistent across all wood species?
- Are there trade-offs between lower moisture content and other properties, such as mechanical strength or brittleness?
Author Response
Comments and Suggestions for Authors (3rd Reviewer)
The overall research is good, but there are some parts need more addressing, please consider the following parts.
- Did you monitor drying conditions (e.g., temperature, relative humidity), or just ambient air-drying?
Answer:
Thank you for the comment. During preparation, we dried the wood specimens in an electrical oven «Mammert» at 60°C. The moisture content of the wood was monitored using a moiture meter.
- What purity grades were used for styrene monomer and potassium peroxydisulfate?
Answer:
Thank you for the comment. Both compounds have a purity of ≥ 99%.
- Why was 600 mmHg vacuum chosen? Was this based on preliminary trials or a specific standard?
Answer:
Thank you for the comment. Yes, this vacuum condition referred to the previous work.
- Was 2% w/w determined experimentally, or adopted from prior literature?
Answer:
Thank you for the comment. We used 2% of potassium peroxydisulphate was based on the previous work which used low concentration of heterogenous catalyst under 5%.
- Did you control humidity during the 24 h polymerization step?
Answer:
Thank you for the comment. We did not control the humidity of the samples, as the polymerization of styrene monomer did not cause water evaporation. The water loss occurred from the moisture content of the wood prior to impregnation.
- Which termite species were predominant at the field site?
Answer:
Thank you for the comment. The termite species predominantly invested at the field site were Mastotermes darwiniensis
- Were L*, a*, and b* values measured immediately after treatment or after conditioning?
Answer:
Thank you for the comment. The color measurements of L, a, and b* were carried out after conditioning, which before-after testing.
- Did you also calculate ΔE for all treatments? You could also mention whether the observed ΔE exceeds thresholds for perceptible or unacceptable discoloration.
Answer:
Thank you for the comment. Yes, ΔE was calculated for all treatments and wood species. The ΔE values can be found in Table 1. According to the literature, ΔE >3 is typically considered a color change perceptible to the human eye, while ΔE >12 is often regarded as a significant change that may be aesthetically unacceptable (Hrčková et al., 2018). In this study, polystyrene-treated wood exhibited an average ΔE of approximately 2–3 (only slightly noticeable), whereas CCA-treated wood showed an average ΔE >20, indicating a very pronounced color change that would likely be considered undesirable for applications emphasizing a natural appearance.
- You may wish to briefly explain why increased density can be beneficial (e.g., improved mechanical performance, durability).
Answer:
Thank you for the comment. Higher wood density is generally positively correlated with increased mechanical strength (e.g., modulus of elasticity and bending strength) as well as enhanced resistance to biological degradation and insect attack. Therefore, the increase in density resulting from polystyrene impregnation can reinforce dimensional stability, wear resistance, and the mechanical load-bearing capacity of wood in end-use applications (Rowell, 2012).
- Besides qualitative mapping, did you quantify the carbon content increase (e.g., atomic percent) to estimate polymer loading more precisely?
Thank you for the comment. In this study, EDS was primarily used for qualitative elemental mapping to confirm the presence of polystyrene in the treated samples. Quantitative analysis of elemental content, such as carbon atomic percent, was not conducted. However, we recognize that such analysis could provide valuable insights into polymer loading, and we consider it a worthwhile direction for future investigation..
- Do you have cross-sectional images showing how deeply the polystyrene penetrated into the wood matrix?
Thank you for the comment. While we did not include detailed cross-sectional images in this study, some supplementary images were taken for qualitative observation. However, due to the relatively low polymer loading in Anisoptera samples, visual differentiation between treated and untreated wood in the cross-sections was minimal. Future work is planned to incorporate more precise imaging techniques, such as SEM or staining methods, to better visualize polymer penetration.
- How did you control for potential artifacts (e.g., beam damage, outgassing of monomer residues) during imaging?
Thank you for the comment. To minimize potential artifacts during SEM-EDS imaging, such as beam damage and outgassing of residual monomer, all samples, both untreated and polystyrene-impregnated, were fully oven-dried before analysis to ensure the removal of any volatile residues. The imaging was conducted at low to moderate accelerating voltage settings to minimize beam-induced damage, particularly in the polymer-treated samples. Additionally, all specimens were gold-coated to enhance conductivity and minimize charging effects. These precautions helped maintain image quality and ensured reliable EDS elemental detection across both untreated and treated samples.
- How do you expect the observed reduction in porosity to affect moisture uptake, swelling, or dimensional stability during service?
Answer:
Thank you for the comment. The reduction in porosity due to polystyrene impregnation is expected to significantly decrease water and moisture absorption because the void spaces within the wood structure are sealed by the hydrophobic polymer. This enhances dimensional stability, reduces swelling and shrinkage during wet–dry cycles, and lowers the risk of damage caused by fluctuations in environmental humidity (Mai & Militz, 2004; Hill, 2006).
- Is there a similar minimum standard for polystyrene-treated wood retention or loading in Indonesia or other countries?
Answer:
Thank you for the comment. At present, there is no national or international standard that specifically establishes a minimum threshold for polystyrene retention in wood, in contrast to CCA preservative retention standards (e.g., SNI or AWPA). However, research by Hadi et al. (1998) demonstrated that a weight percent gain (WPG) exceeding 88% confers high resistance to long-term testing against drywood termites and subterranean termites.
- How do the retention levels observed in CCA-treated samples compare to performance benchmarks for decay or termite resistance?
Answer:
Thank you for the comment. The average CCA retention in this study was 8.4 kg/m³. For comparison, the Indonesian Standard SNI 03-5010.0-1999 specifies a minimum retention of 8 kg/m³ for wood preservation in high-risk applications. Therefore, the CCA retention achieved in this study meets that standard (SNI 1999).
Ref:
[BSN] Badan Standarisasi Nasional. 1999. 03-5010.1-1999. Pengawetan Kayu
Untuk Perumahan dan Gedung. Jakarta (ID): Badan Standarisasi Nasional.
- Was moisture content controlled before treatment, and did it influence polymer uptake?
Answer:
Thank you for the comment. Yes, the moisture content was controlled prior to treatment, as it can indeed influence polymer uptake. By standardizing the moisture content before impregnation, we aimed to minimize variability among samples. This control also allows for appropriate corrections or comparisons at a later stage, ensuring more reliable interpretation of the polymer loading results.
- Was the reduction in moisture content for polystyrene-treated samples consistent across all wood species?
Answer:
Thank you for the comment. Yes, the data in Table 4 show that the reduction in moisture content resulting from polystyrene impregnation is consistent across wood species, ranging from 9.2–9.9%, which is substantially lower than the moisture content of untreated wood (14–17%).
- Are there trade-offs between lower moisture content and other properties, such as mechanical strength or brittleness?
Answer:
Thank you for the comment. The reduction in moisture content and porosity can improve dimensional stability and biological durability. However, polystyrene impregnation may also make the wood stiffer and potentially increase brittleness under impact loads or bending deformation. Consequently, while compressive strength and modulus of elasticity generally improve with increased density, toughness properties may be slightly reduced, which should be taken into account for certain structural applications (Mai & Militz, 2004; Gindl & Gupta, 2002).

Round 2
Reviewer 1 Report
Comments and Suggestions for Authors
Authors are followed the reviewer recommendation.